# Mitochondrial Control Region Database of Hungarian Fallow Deer (*Dama dama*) Populations for Forensic Use

**DOI:** 10.3390/ani14131911

**Published:** 2024-06-28

**Authors:** Orsolya K. Zorkóczy, Zsombor Wagenhoffer, Pál Lehotzky, Zsolt Pádár, Petra Zenke

**Affiliations:** 1Department of Animal Breeding and Genetics, Institute for Animal Breeding, Nutrition and Laboratory Animal Science, University of Veterinary Medicine Budapest, H-1078 Budapest, Hungary; zorkoczy.orsolya.krisztina@univet.hu (O.K.Z.); wagenhoffer.zsombor@univet.hu (Z.W.); 2Hungarian Hunters’ National Chamber, H-1027 Budapest, Hungary; lehotzkyfamily@gmail.com; 3Department of Criminal Sciences, Ferenc Deák Faculty of Law and Political Sciences, University of Győr, H-9026 Győr, Hungary; padar.zsolt@sze.hu

**Keywords:** wildlife crime, forensic genetics, mitochondrial DNA (mtDNA), control region (CR), fallow deer (*Dama dama*), random match probability, population study

## Abstract

**Simple Summary:**

Hungary is world-famous for its fallow deer (*Dama dama*) population and hunting, with approximately 60% of the best trophies originating from this country. Unfortunately, the species also falls victim to poaching. Several studies have already assessed the genetic relationship between fallow deer in certain areas of Europe. The identification of biological materials through mitochondrial DNA analysis has become increasingly important in forensic cases, as it can provide associative evidence connecting victims and suspects. In this study, we determined the extent to which fallow deer mitochondrial control region haplotypes occurring in a given country can be used in legal cases for the preliminary selection of evidence or to link the incriminated animals and the degraded biological remains to the given area. Additionally, we determined which segment of the control region, with its adequate length, should be examined in the case of a genetically narrowed population.

**Abstract:**

The evidential value of an mtDNA match between biological remains and their potential donor is determined by the random match probability of the haplotype. This probability is based on the haplotype’s population frequency estimate. Consequently, implementing a population study representative of the population relevant to a forensic case is vital to correctly evaluating the evidence. The emerging number of poaching cases and the limited availability of such data emphasizes the need for an improved fallow deer mtDNA population databank for forensic purposes, including targeting the entire mitochondrial control region. By sequencing a 945-base-pair-long segment of the mitochondrial control region in 138 animals from five populations in Hungary, we found four different haplotypes, including one which had not yet been described. Our results, supplemented with data already available from previous research, do not support the possibility of determining the population of origin, although some patterns of geographical separation can be distinguished. Estimates of molecular diversity indicate similarly low mtDNA diversity (Hd = 0.565 and π = 0.002) compared to data from other countries. The calculated random match probability of 0.547 shows a high probability of coincidence and, therefore, a limited capacity for exclusion. Our results indicate that despite the overall low genetic diversity of mtDNA within the Hungarian fallow deer samples, a pattern of differentiation among the regions is present, which can have relevance from a forensic point of view.

## 1. Introduction

The fallow deer (*Dama dama*) has a cosmopolitan distribution across nearly every continent worldwide, facilitated by human intervention, and boasts a significant population of more than 40,000 individuals in Hungary [1,2]. Almost 45% of this population is harvested annually, underscoring the species’ substantial game management value. This is due to the infrastructure established to support their hunting, as well as the meat (venison) and trophies obtained from hunting, which also hold cultural and nature conservation significance [3]. Although the hunting of fallow deer in Hungary operates within the legal framework (e.g., Act LV of 1996) [4], instances of poaching still occur. Hunting is only permitted for authorized individuals using approved hunting tools (such as bullet firearms exceeding the energy content established by law) and strictly during the designated hunting season, which differs according to sex and age (Appendix A) [4]. Failure to meet any of these conditions constitutes illegal hunting, a trend highlighted by several studies conducted in Hungary [5,6].

In cases with significant legal implications, proving the suspect’s guilt poses a challenge. However, genetic identification can be employed to compare biological remains (e.g., hairs, blood contamination) found at the scene with those on the suspect’s belongings, as well as identification and selection of the out-portioned, vast number of uncontrolled or illegal meats from private or restaurant freezers. If the test results reveal differences between evidential and reference samples, the genetic evidence cannot support the suspect’s involvement in the crime. Conversely, if there is a match, it strengthens this assumption. The representativeness of the reference samples in the database is crucial in assessing the rarity of DNA profiles associated with evidence from a crime scene when both reference and evidentiary samples originate from the same geographic populations. Hence, a more precise understanding of the genetic structure of local and global deer populations is necessary to interpret matching DNA haplotypes or genotypes accurately. While a genetic method capable of identifying individual fallow deer is available, it is not universally applicable [7]. Many crime scenes are outdoors, leading to the environmental breakdown of evidential samples and significant DNA degradation over time due to factors such as UV exposure, moisture, and bacterial activity.

The advantage of mitochondrial DNA (mtDNA), with its circular structure, is that it is surrounded by a double phospholipid membrane and can be found in multiple copies per cell; thus, it can usually be successfully detected even in small amounts of degraded samples [8]. Therefore, mitochondrial DNA is useful to assist in the identification of the source of a biological sample (such as species and subspecies determination) or to confirm matrilineal relatedness in phylogenetic studies [9,10,11,12,13]. The examination of mtDNA can be important for differentiating populations [14,15] and for determining their geographical origin, as a haplotype may become fixed in a particular region, and the place of origin of an individual can be determined based on the haplotype of a biological sample of unknown origin [9]. The mutation rate of the mitochondrial non-coding control region (CR or D-loop) is five to ten times higher than the average rate of synonymous substitutions of nuclear genes, and therefore more polymorphic, making it widely researched in various Cervidae species [14,15,16,17,18,19,20,21]. Although mitochondrial DNA is not a tool for individualization purposes, owing to its matrilineal inheritance, mtDNA directly links maternal relatives, which can be used as match references where two or more nucleotide discrepancies are needed for a mismatch or exclusion [22], and it is capable of excluding many potential sources despite lower discriminating power than nuclear DNA. For this marker type, it is also necessary to create databases to assess the frequency of alleles or haplotypes within a relevant population [23]. For haploid markers such as mtDNA, where profiles are expected to be shared by many matrilineally related individuals, the strength of the evidence is determined not only by the variability of the sequence but also by the size of the geographically relevant genetic database, which should be large enough to accurately reflect the local diversity [24]. It is well known from previous studies that, especially in fallow deer, the founder effect and the relatively low mutation rate of the mitochondrial genome (compared to microsatellites) indicate that there can be large sets of matrilineally related individuals sharing a common mitogenome [9,10,11,12]. While the CR is widely used in various fields, the primers, and consequently the length of the amplicon, are not standardized. Although shorter sequences are easier to amplify, information loss can occur; therefore, we planned to assess the most informative region. As many articles support the fact that polymorphic sites may occur outside of the conventionally examined, shorter, ‘mutation hotspot’ section, we also aimed to examine the sequence of the entire control region in the samples.

Assessment of the diversity of fallow deer populations based on the mitochondrial CR has already begun in the peripheral regions of Hungary. However, these studies examined only shorter sections of the control region, 708 bp in Southern Hungary (n = 13) and 450 bp in Northeastern Hungary (n = 41) [9,10]. No such data are available from other parts of the country. It is important to determine to what extent the control region can be both sufficient and efficient for the regional differentiation of the domestic fallow deer populations, and thus, to what extent it can assist in cases with legal consequences. For this reason, based on the aggregated (newly defined plus existing) haplotype data, we assessed the probability of matching and whether this extended section of the CR is suitable for the regional separation of domestic fallow deer herds. 

## 2. Materials and Methods

### 2.1. Sampling and DNA Extraction

Muscle or hide samples (n = 138) were collected from registered shootings by hunters with a license between 2019 and 2024 from five regions in Hungary, focusing on those regions in which investigations had not been carried out so far and where the occurrence and hunting of fallow deer are common (i.e., north-west (NW), n = 34; south-west (SW), n = 30; north-middle (NM), n = 42; south-middle (SM), n = 31; and south-east (SE), n = 1). Genomic DNA was isolated using a FavorPrep^TM^ Tissue Genomic DNA Extraction Mini Kit (Favorgen Biotech, Ping-Tung, Taiwan) following the procedural guidelines. The quality of the extracted DNA was tested using 1% agarose gel stained with GelRed^TM^ Nucleic Acid Gel Stain (Biotium, Fremont, CA, USA), and the concentration was measured using a Qubit 2.0 Fluorometer (Life Technologies Corporation, Carlsbad, CA, USA). Isolated DNA from the tissue samples was stored at −20 °C until subsequent analysis.

### 2.2. Mitochondrial Control Region Amplification, Sequencing, and Haplotype Determination

The most informative length and variable sites for the fallow deer CR sequences were determined by an analysis of previous studies [9,10,11,12]. Based on the mitogenome sequence found in GenBank (Accession Number: NC_020700), the entire mitochondrial control region (from 15,400 to 16,146 base pairs) was amplified using newly designed primers (Primer Designer 4 software [http://www.scied.com, accessed on 14 February 2023]) positioned outside the CR. A primer naming convention was used, where the primer name indicates the position of the 5′ base. The forward primer (5′-ACCCCACTATCAACACCC-3′) was defined as F15,386, and the reverse primer (5′-TATGCATAATTAGAGAAAAATTGG-3′) was defined as R16,330.

Amplification was performed in a 25 μL reaction volume containing 12.5 μL DreamTaq™ Green DNA Polymerase (Thermo Fisher Scientific, Waltham, MA, USA), 0.5 μmol forward and reverse primer, 1–10 ng DNA template, and PCR grade H_2_O to volume. PCR was carried out on 2720 Thermal Cyclers (Applied Biosystems, Waltham, MA, USA) using the following conditions: initial denaturation for 10 s at 94 °C, 36 cycles of 40 s at 94 °C, 40 s at 56 °C, 60 s at 72 °C, and a final extension for 2 m at 60 °C. Sequencing reactions of the purified DNA fragments (GenElute™ PCR Clean-Up Kit, Sigma-Aldrich, St. Louis, MO, USA) were carried out with the BigDye^®^ Terminator v3.1 Cycle Sequencing Kit (Thermo Fisher Scientific, Waltham, MA, USA) and on an ABI3500 genetic analyzer (ThermoFisher Scientific, Waltham, MA, USA). Sequence data were analyzed by Sequence Analysis 3.4.1 (Applied Biosystems) and aligned against a reference sequence (GenBank Acc. No: NC_020700) by Sequencher^TM^ 5.4.6 software (Gene Codes Corp, Ann Arbor, MI, USA) for unique haplotype identification. 

Our results were supplemented with the previous control region sequences of 54 Hungarian fallow deer [9,10], downloaded from the NCBI (National Center for Biotechnology Information, Bethesda, MD USA) GenBank database. The downloaded sequences (n = 54) and our new sequences (n = 138) were aligned using MEGA11 software with ClustalW default settings [25]. Statistical tests were performed with a combined examination of the total of these 192 sequences. These sequences were also analyzed separately as six populations mainly divided based on natural barriers such as rivers and lakes.

Wright’s F-statistic was calculated using the DNA Sequence Polymorphism (DnaSP) software [26], and the Fst value was calculated per population pair. Additionally, we determined the number of polymorphic sites and haplotypes found in the sequence for each sampling site and all samples, as well as the haplotype and nucleotide diversity. To estimate the overall haplotype match probability (or random match probability, RMP) for fallow deer sampled at random within the Hungarian population, we used RMP = ∑pi^2^, where p is the frequency of the observed haplotype. The RMP, or probability of matching (PM), is defined as the probability of observing a random match between two unrelated individuals [27] and conveying the significance of a statistical match between reference and evidential sequences in forensic cases.

## 3. Results

Based on previous research and sequences downloaded from GenBank, we determined the investigated section of the control region and the frequency of polymorphic sites in fallow deer introduced to different countries. Sequences from original populations, such as Rhodes or Turkey, were not included in this examination, as they have many more variable sites. This indicates that they still possess some of the original genetic variety of fallow deer not found in the current introduced populations, thus excluding them from the study’s main target. Figure 1 shows that the entire length of the mitochondrial control region, as well as the section that follows, contains variable SNP (Single Nucleotide Polymorphism) sites that contribute to the formation of different haplotypes (see Appendix A for details). Based on this information, a 945-bp-long section containing the whole CR was amplified with the designed primer pair, of which a sequence with a length of approx. 900 base pairs could be reliably evaluated. Due to our selected CR section, four haplotypes were detected for a total of 138 fallow deer samples from the five sampling sites, three of which were already described in previous research [9,10]. A new haplotype sequence was detected from the NM region (Figure 1 and Figure 2), which has been uploaded to GenBank (Accession Number: PP558272). Based on the sequence alignments, we standardized the names of the haplotypes for the current and previous results obtained from Hungarian fallow deer (Figure 1), and thenceforth, we used these names.

We analyzed the observed haplotypes and their frequencies, supplemented with domestic fallow deer sequences (n = 54) from previous research [9,10]. Figure 2 represents the six populations created, within each of which three haplotypes were detected, except for the north-west (NW) region, where only two haplotypes occurred. Altogether, six haplotypes were identified, including six variable sites among the 192 individuals. All polymorphic positions were the result of substitutions, with no indels observed (see Appendix A for more details). Regarding haplotype frequencies among the 192 Hungarian samples examined so far (Figure 2), 66 individuals had Hun1 (34%, previously named H1 or Hap17 [9,10]), and 108 had Hun2 (56%, previously named H2 [9]). One individual each had Hun3 (previously named H3 [9]) and Hun4 (0.5% each). In comparison, nine individuals had Hun5 (4.7%, previously named Hap5), and seven individuals had Hun6 (3.6%, previously named Hap9). Indices of molecular diversity for each group are presented in Table 1. Based on our calculation in the combined dataset, F statistical tests showed that only two southern regions (SM and SE) differ significantly (Fst values ranged between 0.177 and 0.359) from all other sampling locations in Hungary. However, the corrected Gst value, which also considers the population sizes, showed a significant difference in the comparison of only four population pairs (SE-NW, SE-NE, SE-SM, and SM-SW). No significant differences (0.15–0.25) were observed either in Fst or in Gst values between the other sampling sites, and the negative value (effectively seen as zero) calculated between NM and NE indicates that there is no genetic subdivision among these populations. 

The aggregated frequencies from recent and previous studies provided a random match probability between the six areas ranging from 0.429 (SE) to 0.677 (SM). 

## 4. Discussion

The importance or legal status of wild animal representatives can vary by country. Fallow deer from Hungary have some of the finest known antlers of the species, and this is one of the reasons why the species frequently falls victim to poaching. In forensic cases, there is a need to obtain the most accurate and informative data possible. Based on the comparative analysis of fallow deer sequence data from several countries, differences were found in the locations of the polymorphic positions of the mitochondrial control region in introduced populations with very low diversity, due to the founder effect. The number and location of these variations strongly depends on the region (country) of origin of the sequences. This means that sequence regions showing mutational hotspots in one country show no diversity at all in another geographic area. At the same time, it can also be seen that the pattern of sections frequented by mutations is similar in sequences with different origins. The length and position of the corresponding genetic section to be examined must therefore be assessed based on the population study of the given country.

Mitochondrial control region databases are highly valuable for the analysis of minimal amounts of degraded samples (such as hair, feces, and processed samples). These databases have been developed for several species, besides humans, for forensic use [28,29,30,31,32]. For this reason, we aimed to assess the accuracy of the CR sequences of different lengths and whether the implementation of longer sections would be beneficial for forensic investigations. With the help of the 127 fallow deer sequences worldwide, downloaded from GenBank, we showed that mutation hot spots can occur over the entire length of the CR. The examination of the Hungarian fallow deer population revealed that mutations within the CR are concentrated in a shorter section, resulting in all six SNP positions being found within the shorter (450 bp) region previously examined in Hungary [10]. However, further surveys in herds from other countries can result in different patterns.

Because fallow deer, except for the native populations in Turkey and Rhodes, have a very limited genetic diversity [9,10,11,12], every opportunity should be explored to examine potential polymorphic sites, thus detecting possible deviations. The variation reported herein was compared with the published fallow deer genetic data to determine whether our database was typical of fallow deer populations in other countries. Haplotype diversity (Hd) is the probability that the haplotypes of two randomly selected individuals differ, while nucleotide diversity (π) is the average number of nucleotide differences between sequences per base position [33]. The investigated Hungarian fallow deer populations show similarly low average mtDNA diversity (Hd = 0.565) and nucleotide diversity (π = 0.002), compared with data from other countries (Hd = 0–0.902; π = 0–0.01029) [9,11] and previous data from Hungary [10].

While previous studies rarely examined more than 60 individuals per country [9,10], the current study includes broader sampling within the country to uncover additional variation. In our study, a new haplotype was registered; therefore, six haplotypes have altogether been detected so far in Hungary from 192 samples. A similar degree of polymorphism was detected in Germany, where 10 haplotypes were detected by surveying a similarly large number of samples (n = 365) [11]. In other European countries, the number of haplotypes ranged between three and fifteen, investigated from a few dozen samples. The determination of when sufficient samples have been ascertained to adequately represent a population depends on the population size and the mitotype diversity observed within a population. An ideal sample set would be considered saturated when sampling additional individuals from the population no longer increases the absolute number of observed types [34]. Generally, the number of observed haplotypes increases with sample size, while the proportion of rare haplotypes (i.e., encountered only once or twice) decreases [35]. Our results in most of the populations investigated support these statements; as a result, the exclusion probability largely remained the same with sample size expansion [36]. In one of the regions investigated in this study, only a limited sample size was available (SW, n = 7), thus causing a tendency to overestimate haplotype frequencies, which decreases the evidential value of an mtDNA match. Because overestimations do not increase the risk of incriminating a false suspect, under-sampling can be considered a conservative error [37].

In general, genetic subdivisions and inbreeding must be considered in wildlife forensic DNA analysis, particularly in the case of fallow deer. DNA databases that reflect the genetic composition and geographic structure are important for accurately calculating the rarity of allele frequencies and mtDNA haplotypes to determine exclusion probabilities [22]. Therefore, we examined the possibilities of the mitochondrial DNA control region to determine whether there is geographical separation of the haplotypes, and this can also help solve cases with legal consequences. To evaluate the significance of a haplotype match between a biological trace and its suspected donor, a population sample should reliably represent the population to which the donor of the trace is supposed to belong. Regarding the distribution of domestic haplotypes, Hun3 and Hun4 have, so far, only occurred in one individual each in the north-eastern (NE) and north-middle (NM) regions of Hungary, respectively; Hun5 and Hun6 only occur in Southern Hungary, in nine and seven individuals, respectively. The other two haplotypes (Hun1 and Hun2) were far more common at 90.6%, with these occurring almost throughout the entire country. Although earlier genetic research conducted in European fallow deer showed a high degree of nuclear and mitochondrial DNA diversity, only a small degree of variation is present per country due to the fixation of local allelic variants [9,10,14,19,38,39,40,41,42,43,44]. The Fst (fixation index) and Gst values provide information on the segmentation of the populations, as it shows how much the proportion of haplotypes decreases because the metapopulation consists of two or more subpopulations with different haplotype frequencies. The higher the fixation index, the more suitable the genetic marker is for separating populations. Fst ≥ 0.15 already indicates a significant genetic difference between subpopulations [45]. Our results indicate that despite the overall low genetic mtDNA diversity within the Hungarian fallow deer samples, a pattern of differentiation among the regions is present, which can have relevance from a forensic point of view.

We are aware of the limitations of our dataset in identifying individuals derived from the study of autosomal microsatellites [7]; however, these tests can be conducted faster and more easily. Furthermore, in case a nuclear DNA test is unsuccessful, this type of extranuclear genome can likely be primarily used as an exclusionary tool. Nonetheless, using these techniques without reference data for comparison may lead to incompatible case reporting; therefore, comprehensive domestic genetic databases are needed. These can help investigators trace samples to their sources of origin (population, species, geographic region) and aid in the arrest, conviction, and subsequent sentencing of perpetrators for smuggling, poaching, or possession [46]. Despite the fact that our calculated random match probability (RMP) of 0.547 shows a high probability of coincidence and, therefore, a limited capacity of exclusionary potential, this work will help to interpret the strength of evidence in forensic cases. As the power of genetic evidence is directly correlated to the exclusionary power generated by the local haplotype frequencies available [34], this dataset provides an additional forensic asset for a fallow deer mtDNA CR database.

Beyond forensic considerations, mtDNA data are indispensable tools in population management and conservation. They provide insights into genetic diversity, population structure, phylogeography, and lineage tracking. These insights help conservationists make informed decisions to restore healthy, genetically diverse populations, thereby enhancing the long-term viability of species.

## 5. Conclusions

Country-by-country differences in the location of polymorphic positions of the mtDNA control region necessitate the assessment of local populations in connection with the examination of the most informative sections. By increasing the number of tested sample elements, although it is possible to detect new haplotypes, the probability of exclusion remains largely the same. Based on the detected uneven haplotype frequencies, there can be a partial geographical separation between the surveyed stocks. However, the probability of random matching within the populations is still very high in Hungary. Consequently, it may be necessary to search for polymorphic sites on other mitochondrial gene sections or to sequence the entire mitogenome of fallow deer.

## Figures and Tables

**Figure 1 animals-14-01911-f001:**
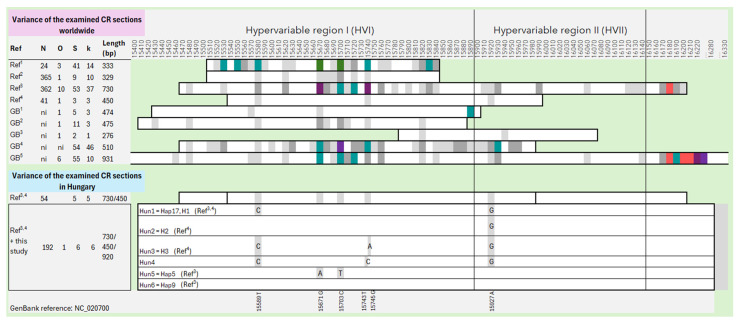
Comparison of the polymorphic sites of the mitochondrial control region (CR) to the reference sequence (NC_020700) based on the length of the investigated segment in introduced fallow deer stocks. Ref: previous studies and sequences downloaded from GenBank (GB). Color codes: 
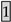


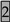


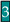


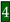


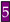


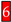
 polymorphic sites per 10 base pairs. N: number of sequences. O: number of countries the samples originated from. S: number of variable sites. k: number of haplotypes. Ref^1^: [12]; Ref^2^: [11]; Ref^3^: [9]; Ref^4^: [10]; GB^1^: MK473448-50; GB^2^: ON321840-41; GB^3^: OQ535577-79; GB^4^: OR220344-89; GB^5^: OR232305-17; ni: no information.

**Figure 2 animals-14-01911-f002:**
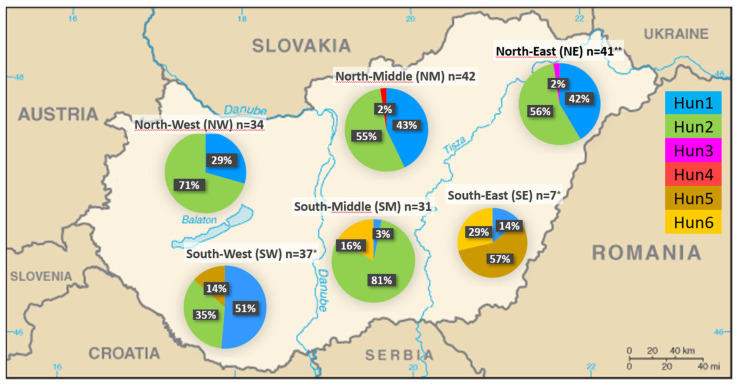
The distribution of haplotypes Hun1–Hun6 of the fallow deer populations studied in Hungary in six regions. n: sample size. *: seven and six sequences from populations SW and SE, respectively, originated from a previous study [9], **: sequence data originated from a previous study [10].

**Table 1 animals-14-01911-t001:** Parameters of the six sampling sites analyzed by the DnaSP program. NW = North-West, NM = North-Middle, NE = North-East, SW = South-West, SM = South-Middle, and SE = South-East. n: sample size. S: number of variable sites. k: number of haplotypes. Hd: haplotype diversity. π: nucleotide diversity. RMP: random match probability. Fst: fixation index. Gst: coefficient of gene differentiation (with blue color). *p*: probability value.

Region	NW	NM	NE [10]	SW	SM	SE	Overall
**n**	34	42	41	37	31	7	192
**S**	1	2	2	3	2	4	6
**k**	2	3	3	3	3	3	6
**Hd**	0.428	0.529	0.526	0.611	0.333	0.667	0.565
**π**	0.00095	0.00123	0.00123	0.00221	0.00076	0.00423	0.00162
**RMP**	0.585	0.484	0.487	0.617	0.677	0.429	0.547
**NW**		0.025	0.017	0.084	0.177	0.345	**Fst**
**NM**	0.010		−0.022	0.026	0.298	0.359	(*p* < 0.001)
**NE**	0.006	−0.012		0.028	0.286	0.356	
**SW**	0.072	0.015	0.019		0.291	0.237	
**SM**	0.050	0.111	0.104	0.189		0.324	**Gst**
**SE**	0.211	0.148	0.151	0.100	0.265		(*p* < 0.001)

## Data Availability

The data will be available from the corresponding author upon request.

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
