# Peer review of "Mitochondrial Control Region Database of Hungarian Fallow Deer (Dama dama) Populations for Forensic Use"

_animals, 2024, doi:10.3390/ani14131911_

Round 1

Reviewer 1 Report

Comments and Suggestions for Authors

The authors in this manuscript sequenced the control region of the Dama dama populations from Hungary and speculated that the data could be useful for forensics. The manuscript is well written and could be interesting to the readers of Animals. There are some limitations in the study which are to be fixed before publishing. The major limitation is the sample size. They have sequenced 138 samples from six localities which is itself a small number to represent the entire Hungary. I noticed the authors acknowledged the limitations in the last paragraph of the discussion, but still, some of the limitations could be overcome.

Increasing the sample size might not be feasible at this stage, hence, the authors should focus on analyzing those data in multiple possible ways and drawing conclusions. I have provided some suggestions in the annotated PDF and a few major comments are as follows:

Introduction: Line 54-55: Please mention specifically what type of tools are approved for hunting and which specific seasons. 

Materials and methods: line 119: Specify the sampling locations here

Line 141: Initial denaturation step is missing here?

Data analysis:

The authors are suggested to run some additional analysis. 

-You have different sample sizes from different localities, you should test statistically if there is effect of sample size in number of haplotypes. 

- Haplotype network showing the mutational relationship among the haplotypes would be informative. 

- Isolation by distance phenomenon could be tested using Mantel test. 

- Another major objective of the research was to test whether using the entire D loop yields more haplotypes than using HVR1 or HVR2 segments alone. It could have been tested statistically.

Figures and tables are cited repeatedly and figure 2 comes before figure 1 in the text. Please fix.

How long was the sequence of CR (in bp) you retrieved? It has never been mentioned in the manuscript, neither in methods nor in the results in the text. 

In the Figure 1, the term CR variance worldwide might be inappropriate. For example, sequences from [10], [11] are for HVR1 only. Those studies might have analyzed only HVR1 fragment. Does it mean that the Dama dama from those countries have only HVR1 in the CR? It needs to be clearly presented. The authors might need to improve the presentation of the Figure 1. 

The results section contains some contents of discussion in which authors have cited literature. 

The discussion is basically well-written. However, after running the above-recommended analysis, it needs to be restructured accordingly.

References have several issues. 

All the best!

Author Response

Author's Response to Reviewer Comments

First of all, we would like to thank the Referees for their time and effort spent with our manuscript.

Our detailed replies to their specific requests and comments are as follows (references to locations in our responses below are to the “Tracked Changes” version of the revised manuscript, which is attached to the resubmitted version).

Response to Reviewer 1 Comments

“Line 54-55: Please mention specifically what type of tools are approved for hunting and which specific seasons.“

Response: Added the requested information to the manuscript (lines 53-57).

“Materials and methods: line 119: Specify the sampling locations here”

Response: Added the requested information to the manuscript (lines 122-123).

“Line 141: Initial denaturation step is missing here?”

Response: We added the requested information to the manuscript (lines 143-144).

“The authors are suggested to run some additional analysis.

  • You have different sample sizes from different localities, you should test statistically if there is effect of sample size in number of haplotypes.” 

Response: We thank the Reviewer for pointing out this problem. Therefore, we included the Gst statistic results in Table 1 and in the manuscript (lines 212-216).

  • “Haplotype network showing the mutational relationship among the haplotypes would be informative.” 

Response: We thank the Reviewer for the idea and included the requested statistic (Supplement Figure S1) in the manuscript (lines 203-204).

  • “Isolation by distance phenomenon could be tested using Mantel test.” 

Response: Although the Mantel test helps to draw meaningful conclusions about the spatial and environmental factors influencing haplotype variation, according to the authors, its forensic significance is not so pronounced. Therefore, the authors would like to ignore this request.

  • “Another major objective of the research was to test whether using the entire D loop yields more haplotypes than using HVR1 or HVR2 segments alone. It could have been tested statistically.”

Response: We thank the Reviewer for pointing out this possibility. The number of mutation sites and haplotypes obtained by examining the HV1 region was compared with the number of mutation sites and haplotypes obtained by reviewing the entire control region (HV1+ HV2). The results have now been included in the Supplement material (Table S2-S3) and referred to in the text (lines 178-180).

“Figures and tables are cited repeatedly and figure 2 comes before figure 1 in the text. Please fix.”

Response: Thank you for bringing this error to our attention. We have corrected this issue.

“How long was the sequence of CR (in bp) you retrieved? It has never been mentioned in the manuscript, neither in methods nor in the results in the text.” 

Response: We thank the Reviewer for drawing attention to this shortcoming. Therefore, the exact amplicon size and the length of the reliably evaluable sequence obtained from it have been included in the manuscript (lines: 31-32, and 181-183).

“In the Figure 1, the term CR variance worldwide might be inappropriate. For example, sequences from [10], [11] are for HVR1 only. Those studies might have analyzed only HVR1 fragment. Does it mean that the Dama dama from those countries have only HVR1 in the CR? It needs to be clearly presented. The authors might need to improve the presentation of the Figure 1.” 

Response: We appreciate the Reviewer's criticism of Figure 1 and have reworded the captions and improved the explanatory text to better convey the intended meaning of this figure. (During the various studies, the regions of the control region with different lengths and locations were examined with different sample numbers. The authors wanted to present these aggregated results in Figure 1, supplemented by the occurrence of the degree of polymorphism experienced in the given sections, and compare them with the Hungarian haplotype variance results performed in the present research.) However, we would like to keep this figure essentially in this form.

“The results section contains some contents of discussion in which authors have cited literature.“

Response: We included these references in the results section because the haplotype results of these (previously performed on fallow deer in Hungary) were aggregated and analyzed uniformly with the results obtained in our present research.

“The discussion is basically well-written. However, after running the above-recommended analysis, it needs to be restructured accordingly.”

Response: We have corrected the discussion according to the results of the incorporated new statistics (lines 305-306).

“References have several issues.” 

Response: We have fixed Latin species names to italic format.

Other modifications requested by the Reviewer (marked in the pdf format of our manuscript) were also transferred to the revised version.

Reviewer 2 Report

Comments and Suggestions for Authors

Comments on the Quality of English Language

The English is good.

Author Response

Author's Response to Reviewer Comments

First of all, we would like to thank the Referees for their time and effort spent with our manuscript.

Our detailed replies to their specific requests and comments are as follows (references to locations in our responses below are to the “Tracked Changes” version of the revised manuscript, which is attached to the resubmitted version).

Response to Reviewer 2 Comments

“In this paper the mtDNA control region (CR) was sequenced from 138 fallow deer in five

populations in Hungary. Four mtDNA haplotypes were identified with DNA sequence

substitutions. Three of the four haplotypes have been reported in other fallow deer populations.

Comparisons with other fallow deer populations’ mtDNA sequences were made. The probability

of a random match has a high probability of coincidence (i.e., different deer have the same

haplotype) making the CR in Hungary limited for positive identification and exclusion of suspect

deer samples compared with subject deer samples in poaching forensic cases.

This is a very well-written paper, with excellent description of methods and results. I offer the

following comments that can improve the paper.”

“1. This paper reports what could be considered negative results, in which the objective of

finding a genetic marker to identify individuals was not achieved. This is good, negative

results are valuable. The robust data in the paper are valuable for forensics as described

below, as well as for population genetics, phylogenetics, and conservation genetics

applications.

  1. The sequence data are valuable for forensics in that different haplotypes in a suspect and

a poached sample would exclude the suspect and prevent an incorrect prosecution. Even

though the probability of individual identification is low, some suspect samples could be

excluded. I think this should be emphasized in the paper.”

Response: Thank you for your positive comment.

“3. The paper could emphasize the utility of the mtDNA data for population management and

conservation. If fallow deer populations have been recovered with introductions, founder

effects, and population bottlenecks, the number of haplotypes and their distribution could

be used as an indicator of populations’ genetic variation and used to design future

translocations to enhance genetic diversity and population fitness.”

Response: We thank the Reviewer for the idea and have added a paragraph emphasizing the utility of the mtDNA data for population management and conservation (lines 328-332).

“4. Lines 70-71, add a reference for identifying individual deer.”

Response: We have added a reference (line 72).

“5. Line 75, describe what the strong protein coat is (e.g, the mitochondrial membrane?).”

Response: We thank the reviewer for drawing attention to this mistake. We have corrected it (line 76).

“6. Lines 83 and 97, clarify, regarding high and low mutation rate; I think the mtDNA

mutation rate is generally greater than the mutation rate for nuclear DNA.”

Response: Thank you for drawing our attention to this contradiction. We have reworded the sentence (lines 98-99).

“7. Line 99, say “PCR primers”.”

Response: The authors don't know how to fix it, please clarify your concerns.

“8. Line 110, describe the domestic fallow deer more and how they relate to the wild, hunted

populations.”

Response: In Hungary, only wild, hunted populations of fallow deer exist. We used the expression domestic fallow deer as a synonym for the Hungarian fallow deer.

“9. Table 1, Describe the lower Fst matrix in the Table legend.”

Response: A description is added regarding the Fst in the Table legend.

“10. Figures 1 and 2 are good, but I think including a simple Table with the populations and

their haplotypes would make the results more clear.”

Response: The Reviewer is right that a simple table is more transparent, but we would not have been able to perform a polymorphism comparison of so many sequences in it. For this reason, we would like to keep both figures, but for the sake of easier understanding, we slightly modified Figure 1. Furthermore, we supplemented the manuscript with a table containing the mutation numbers (Table S2).

“11. Line 206, is the <0.15 an Fst or significance p value?”

Response: A correction has been added to the text (lines 212-216) and Table 1.

„12. Lines, 233-235 and 304-305, Excellent point that could be made in the abstract.”

Response: Thank you for the suggestion. We made the second recommended sentence in the abstract (lines 39-41).

„13. Line 273, make it (SW, N = 7)”

Response: We have made the suggested correction (line 284).

„14. Line 275 and throughout the paper, consider replacing “since” with “because”.”

Response: We have made the suggested replacements (lines 257, and 285).

„15. Discussion, Are there any morphological or reproductive indications of inbreeding

depression?”

Response: In the last decade, a purulent antler inflammation disease has appeared in fallow deer living in our country. The exact cause of this is unknown, but presumably, the deterioration caused by inbreeding makes the herd more sensitive to various infections.

„16. Line 305, good points, see my comment 2 above.”

Response: We made it in the abstract (lines 39-41).

„17. A possible reference regarding the early use of mtDNA in forensics: Cronin et al. 1991.

Mitochondrial DNA in wildlife forensic science: species identification of tissues.

Wildlife Society Bulletin 19:94-105.”

Response: We have added the recommended reference (line 80).

Reviewer 3 Report

Comments and Suggestions for Authors

The authors might consider the following in editing their manuscript:

1)   Figure 1.  Change ‘hipervariable’ to ‘hypervariable’?

2)   In general, how ‘populations’ were defined may impact results quite a bit.  Are the authors confident that dividing Hungary into these six areas is fully justified?  Many of the conclusions are based on the assumption that these areas represent some sort of real biological entities? 

3)   Line 254.  Haplotype and nucleotide diversity may be impacted by the length of the mtDNA sequence region that is sampled.  Does the comparison to other geographic areas here take into account the fact that the mt control region sequenced might not be the same length in the different studies that are being compared?

4)   It would be of interest to show a tree of the various control region haplotypes for this species of deer for all fallow deer control regions that have been sequenced to date.  This would be of interest to show where different haplotypes are found and in what frequency for various countries or populations.  Although the focus in the paper is on Hungarian specimens, this broader perspective might be of interest to the reader and may provide additional insights about where different haplotypes are found and which haplotypes are restricted to just certain populations or countries?

Author Response

Response to Reviewer 3 Comments

The authors might consider the following in editing their manuscript:

  • Figure 1.  Change ‘hipervariable’ to ‘hypervariable’?

Response: We thank the reviewer for drawing attention to this mistake. We have corrected it.

  • In general, how ‘populations’ were defined may impact results quite a bit.  Are the authors confident that dividing Hungary into these six areas is fully justified?  Many of the conclusions are based on the assumption that these areas represent some sort of real biological entities? 

Response: The east-west division was based on the country's two main rivers, as they are natural barriers between the populations. The north-south division was based on the occurrence of fallow deer populations and the difference in habitat types.

  • Line 254.  Haplotype and nucleotide diversity may be impacted by the length of the mtDNA sequence region that is sampled.  Does the comparison to other geographic areas here take into account the fact that the mt control region sequenced might not be the same length in the different studies that are being compared?

Response: The reviewer is correct that these diversity values may be impacted by the length of the mtDNA sequence region that is sampled. The authors are also aware of the problem that the sequences underlying the different studies are indeed not of the same size. However, since they overlap to a large extent, we found it suitable to make the comparison.

  • It would be of interest to show a tree of the various control region haplotypes for this species of deer for all fallow deer control regions that have been sequenced to date.  This would be of interest to show where different haplotypes are found and in what frequency for various countries or populations.  Although the focus in the paper is on Hungarian specimens, this broader perspective might be of interest to the reader and may provide additional insights about where different haplotypes are found and which haplotypes are restricted to just certain populations or countries?

Response: The reviewer is right, such a haplotype network or tree would be of interest to many researchers dealing with fallow deer. However, there are many obstacles to constructing such trees, e. g. the varying lengths of referred sequenced regions used as datasets.  While, based on the comparison of the control region sequences examined so far, there is a short section of overlap in all of them, it is even shorter than the HVI section. Thus, using it as the basis of the analysis would greatly reduce the information content and with it the number of haplotypes. If, on the other hand, we compare them over a larger section, we lose many series of sequence sets due to their brevity.

Round 2

Reviewer 1 Report

Comments and Suggestions for Authors

Thank you for revising the manuscript by incorporating the suggestions. All the best!